# Different Moro Zones of Psoas Major Affect the Clinical Outcomes after Oblique Lumbar Interbody Fusion: A Retrospective Study of 94 Patients

**DOI:** 10.3390/jcm12030989

**Published:** 2023-01-27

**Authors:** Zefeng Song, Xingda Chen, Zelin Zhou, Wanyan Chen, Guangye Zhu, Rueishiuan Jiang, Peng Zhang, Shaohao Lin, Xiaowen Wang, Xiang Yu, Hui Ren, De Liang, Jianchao Cui, Jingjing Tang, Xiaobing Jiang

**Affiliations:** 1First Clinical Medical College, Guangzhou University of Chinese Medicine, Guangzhou 510405, China; 2Department of Spinal Surgery, The First Affiliated Hospital of Guangzhou, University of Chinese Medicine, Guangzhou 510405, China

**Keywords:** Moro zones, minimally invasive, psoas major, oblique lumbar interbody fusion, teardrop-shaped psoas major

## Abstract

Oblique lumbar interbody fusion (OLIF) has been driven to the maturity stage in recent years. However, postoperative symptoms such as thigh paresthesia resulting from intraoperative retraction of the psoas major (PM) have sometimes occurred. The aim of this study was to assess the different positions and morphology of PM muscles and their relationship with clinical outcomes after OLIF by introducing the Moro zones. Patients who underwent L4-5 OLIF at our institution between April 2019 and June 2021 were reviewed and all data were recorded. All patients were grouped by Moro zones into a Moro A cohort and a Moro I and II cohort based on the front edges of their left PM muscles. A total of 94 patients were recruited, including 57 in the Moro A group and 37 in the Moro I and II group. Postoperative thigh pain or numbness occurred in 12 (21.1%) and 2 (5.4%) patients in the Moro A group and the Moro I and II group, respectively. There was no difference in the psoas major transverse diameter (PMTD) between groups preoperatively, while longer PMTD was revealed postoperatively in the Moro A group. The operating window (OW) and psoas major sagittal diameter (PMSD) showed significant differences within and between groups. Thirteen patients had teardrop-shaped PM muscles, with 92.3% in the Moro A group showing significantly worse clinical scores at 1-week follow-up. The Moro zones of the PM affected the short-term outcomes after OLIF. Preoperative measurements and analysis of OW, PMSD and PM morphology should be performed as necessary to predict short-term outcomes.

## 1. Introduction

The effectiveness of posterior lumbar interbody fusion (PLIF) and transforaminal lumbar interbody fusion (TLIF) has been proven. However, their damage to the posterior musculoligamentous complex cannot be avoided, so oblique lateral lumbar interbody fusion (OLIF) has been proposed for the indirect decompression of the spinal canal through a retroperitoneal oblique approach [1,2]. This surgical corridor is a natural anatomic gap between the psoas major (PM) and the abdominal aorta. The gap can be used to implant a higher, wider, and larger cage into the target vertebral space to better restore physiological lumbar lordosis [3]. In contrast to extreme lateral interbody fusion (XLIF) and direct lumbar interbody fusion (DLIF), OLIF does not necessitate the splitting of the psoas major (PM) [4,5,6,7]. Several advantages of OLIF, such as a higher fusion rate and fewer postoperative complications, have been reported [8].

Nevertheless, there is still a risk of nerve and muscle injury during OLIF, with an incidence of up to 35.4% [3,9]. Hip flexion weakness or paresthesia at the anterolateral thigh is thought to be the most common complication. Many scholars believe that the postoperative complications of OLIF are related to the injury of the lumbar plexus, genital femoral nerve, and lateral femoral cutaneous nerve caused by the distraction of the PM during operation [10]. Thicker PM muscles and narrower operating areas are related to a higher degree of intraoperative strain on organs, nerves, and soft tissues. Yet, the correlation between the position and morphology of the PM and these complications has not been reported. Thus, this study introduces the Moro zone and classifies the PM based on the relative location between the PM and the disc, and investigates the relationship between different Moro zones of the PM and clinical scores after OLIF [11,12].

## 2. Materials and Methods

### 2.1. Patients and Procedures

With the approval of the ethical committee, we performed a systematic retrospective analysis of 103 consecutive patients who underwent L4-5 OLIF at our institution between June 2018 and May 2020. Nine cases were lost to follow-up, and ninety-four cases were finally included, with at least 1-year follow-up. Each patient’s vertebra was divided from anterior to posterior edges into Moro Zones A, I, II, III, IV, and P. All patients in our study were grouped into Moro A and Moro I and II cohorts based on the front edges of their left PM muscles using axial magnetic resonance imaging (MRI). Demographics, including gender, age, body mass index (BMI), bone mineral density (BMD), and symptom duration, were recorded.

This study, with independent institutional approval, was performed in line with the principles of the Declaration of Helsinki. All patients gave written informed consent.

### 2.2. Inclusion and Exclusion Criteria

Inclusion criteria consisted of age > 18 years, I° or II° lumbar spondylolisthesis, lumbar instability, mild to moderate spinal stenosis or discogenic lumbago at the L4-5 level, severe lumbar pain with or without lower-limb radicular symptoms, intermittent claudication which could be relieved by rest for more than 70%, and failed conservative treatment for more than 3 months. Those who had giant lumbar disc herniation or prolapse, severe lumbar spinal stenosis, tumor, fracture, severe medical comorbidity or severe osteoporosis, or a history of previous abdominal or retroperitoneal surgery and those who were lost to follow-up or radiographic examinations were excluded.

### 2.3. Outcome Observations

Preoperative and postoperative imaging parameters, including the disc sagittal diameter (DSD), operating window (OW), psoas major sagittal diameter (PMSD), and psoas major transverse diameter (PMTD) in all patients, were recorded based on the method of Chen X [13] (Figure 1 left). Postoperative imaging analysis was performed with MRI 1 week after surgery. Meanwhile, based on the method of Moro, the axial view of the intervertebral disc was divided into six regions, and the location of the left PM was evaluated [11]. Zones I to IV were of equal width. Beyond the anterior and posterior edge of the disc, the two regions were defined as Zone A and Zone P, respectively. The Moro zones of the PM were recorded, depending on whether the PM muscles were “teardrop-shaped” on MRI (Figure 1 right). Complications, estimated blood loss, and the length of hospital stay were noted. The operation time that we recorded did not include posterior percutaneous screw fixation (PPSF) because the PM needed to be retracted only in the oblique corridor. The visual analogue scale (VAS) and Oswestry disability index (ODI) scores were assessed postoperatively at 1 week, 3 months, 6 months, and 12 months.

### 2.4. Surgical Technique

Each patient was placed on the operating table in the right lateral position under general anesthesia. Axillary pads were put underneath the axilla and iliac crest and the abdomen was allowed to hang naturally. After the spine was temporarily fixed, the surgical segment was confirmed by fluoroscopy. The skin and subcutaneous tissues were incised sequentially, and the left external oblique abdominal muscle, internal oblique abdominal muscle, and transverse abdominal muscle were bluntly separated. The abdominal organs and large vessels were compartmentalized anteriorly, and the PM was retracted posteriorly. Subsequently, the intervertebral space was exposed and the venous plexus on the surface of the disc was electrically coagulated. After the insertion of the Kirschner wire, the target segment was reconfirmed by fluoroscopy. Then, the left annulus fibrosus was incised, and the nucleus pulposus and the cartilage endplates were cleaned. After testing with a trial mold, an optimum-sized cage was checked. Filled with allograft bone, the cage was implanted under direct vision, with fluoroscopy confirming a satisfactory position. Eventually, the muscles, subcutaneous tissues, and skin were sutured in layers. The patient was then transferred to the prone position for PPSF. 

### 2.5. Statistical Analysis

SPSS 23.0 (SPSS Inc., Chicago, IL, USA) software was applied for statistical analysis., The chi-square test and Fisher’s exact test were used to compare categorical variables between the two cohorts. The independent-sample t test or Mann–Whitney U test was performed to analyze numerical variables. The paired t test was used to compare clinical and imaging outcomes before and after surgery. Pearson correlation analysis was applied to analyze the correlation between imaging parameters and postoperative clinical scores, and multivariate linear regression analysis was used to analyze the imaging parameters that influenced clinical outcomes after OLIF. A *p* value of <0.05 was considered statistically significant.

## 3. Results

According to our observation, no PM was found to be located in Zones III, IV, and P. There were 57 cases in the Moro A group, including 21 males and 36 females, with a mean age of 60.04 ± 10.03 years. The Moro I and II group included 37 cases, 10 males, and 27 females, with a mean age of 60.68 ± 9.13 years. There were no statistical differences in age, sex, BMI, BMD, symptom duration, background diseases, or main diagnosis between the groups (Table 1).

In terms of perioperative data, there were no statistical differences in the operation time and length of hospital stay between the two groups. However, the Moro A group had more estimated blood loss than the Moro I and II group, and the difference was statistically significant (*p* = 0.043). The overall complication rate was 36.17%, including 12 (12.77%) cases of cage displacement, 6 (6.38%) cases of cage subsidence or endplate injury, and 2 (2.13%) cases of sympathetic chain injury. None of the above complications were statistically different between the two groups. Notably, postoperative thigh pain or numbness and hip flexion weakness occurred in 12 (21.05%) and 2 (5.41%) patients in the two groups, respectively, with statistically significant differences (*p* = 0.042) (Table 2). All complications related to nerve or muscle injury were relieved by oral neurotrophic drugs and NSAIDs. Patients with cage subsidence did not develop severe symptoms during the follow-up period.

The parameter measurements of preoperative DSD and PMTD on MRI at the L4-5 level showed no statistical difference between the two groups. The preoperative PMSD was 25.17 ± 3.92 and 17.51 ± 3.34 mm in the Moro A group and the Moro I and II group, respectively, with statistical significance (*p* < 0.001). The preoperative OW was 15.81 ± 3.74 and 17.45 ± 3.01 mm in each group, respectively, showing statistical significance (*p* = 0.028). Thirteen cases of teardrop-shaped PM were found in two groups, with 92.3% in Moro Zone A (*p* = 0.012). As with preoperative data, the Moro A group had a longer postoperative PMSD (*p* < 0.001) and a narrower postoperative OW (*p* = 0.013) than those of the Moro I and II group. By contrast, the PMTD changed from no difference preoperatively to significantly longer postoperatively (*p* = 0.003). At the same time, the PMTD and PMSD lengthened postoperatively in both groups compared to pre-operation (*p* < 0.001), while the OW narrowed correspondingly (*p* < 0.001), with a greater variation extent in the Moro A cohort (Table 3). 

Clinical scores showed a significant improvement in the VASb, VASl, and ODI scores at all postoperative follow-up times compared to preoperative scores in both groups. It was worth noting that at the 1-week postoperative follow-up, all clinical scores were significantly worse in the Moro A group (*p* < 0.001) (Figure 2). However, clinical scores at other follow-up times revealed no significant difference between the two cohorts. Pearson correlation analysis of the preoperative DSD, OW, PMSD, and PMTD relative to clinical scores showed that the preoperative OW was negatively correlated with ODI scores at 1 week postoperatively (*p* < 0.001). There was a positive correlation between the preoperative PMSD and the 1-week postoperative VASb and VASl (*p* < 0.001), the same positive correlation between the preoperative PMSD and the ODI (*p* = 0.007), and the highest correlation coefficient between the preoperative PMSD and the 1-week postoperative VASl (r = 0.621). A multiple-regression linear model was constructed from the preoperative OW and PMSD, which had a significant linear correlation with the 1-week postoperative ODI (Table 4) (Figure 3).

## 4. Discussion

OLIF has been widely used in spine surgery because it allows back muscles and tissues to be well protected. Consistent with previous findings, our study reconfirmed that OLIF is safe and effective in the treatment of degenerative lumbar diseases [14,15,16,17,18,19]. Intervertebral space, foramen, and ligamentum flavum were distracted intraoperatively, therefore relieving neurological symptoms caused by intervertebral space collapse or foraminal stenosis and dorsal compression of the thecal sac. In addition, a larger-sized cage combined with PPSF was able to effectively relieve low-back pain caused by mechanical instability. Thus, OLIF has gained more attention with reliable clinical efficacy [20,21]. Figure 4 demonstrates a typical case with 1-year follow-up.

Though OLIF avoids damage to the posterior complex structures, it inevitably separates and distracts the abdominal aorta, PM, and lumbar plexus, which increases the risk of retroperitoneal vascular, muscular, and neural injury. This is also thought to cause the sympathetic chain and iliopsoas symptoms [22]. Previous studies have found that the incidence of sensory nerve injury and PM weakness ranges from 10.8% to 35.4% among the postoperative complications of OLIF [3,9,20,23,24,25]. In our study, 34 patients developed complications, and of these 34, 14 cases (14.9%) included thigh pain or numbness and hip flexion weakness, which represents similar incidence to the previous studies. Those symptoms are closely related to intraoperative PM retraction [26,27]. To the best of our knowledge, no study has reported the relationship between the postoperative complications of OLIF and the morphology and position of the PM so far. Therefore, we introduced the Moro zones of the PM to investigate this relationship, aiming to provide predictive factors for short-term complications and clinical outcomes.

In this study, the positions of the left PM muscles were described based on the method proposed by Moro. Cases in Moro Zones III, IV, and P went undetected among the population. There was no statistical difference in demographics, perioperative data, and follow-up outcomes between patients in Moro Zones I and II, so the two cohorts were combined into one group for discussion. Zhe Wang et al. analyzed 300 patients and classified 205 cases (68.3%) into Moro Zone A and 95 cases (31.7%) into Moro Zone I and II at the L4-5 level [28]. Their results are similar to the results of our study (Moro A: Moro I and II = 60.6%: 39.4%), reconfirming the reasonability of this classification. To further investigate the imaging difference between the two groups, we analyzed the DSD, OW, PMSD, and PMTD. On account of the PM needed to be retracted close to the horizontal midline of the vertebra to fully expose the surgical field intraoperatively, we used the disc horizontal midline as the reference line. DSD and PMSD were defined as the distance from the anterior edge of the disc and the left PM to the reference line, respectively, evaluating the thickness of the PM. The distance that the reference line crossed the left PM was defined as the PMTD and used to access the width of the PM. Lines from the disc midpoint to the left border of the abdominal aorta and the anterior border of the left PM, respectively, produced two intersection points at the disc. The distance between the two points was defined as the OW, determining the size of the surgical area. The PMSD in Moro Zone A was longer than that in Moro Zone I and II preoperatively because the PM muscles in Zone A were closer to the ventral side. A longer PMSD would shorten the distance between the PM and the abdominal aorta, shrink the OW, and compress the surgical area, so that the operative corridor was naturally narrower, meaning that intraoperative complications were more likely to occur.

PM swelling might affect clinical efficacy. Chen X et al. found that operating windows were largest at L2-3, followed by L3-4, L5-S1, and L4-5, by analyzing 400 patients who had undergone OLIF [13]. In addition, the left annulus fibrosus was covered by the PM normally. Therefore, it was inevitable that performing OLIF at the L4-5 level resulted in the retraction of the PM, especially when clearing the nucleus pulposus, which meant that PM swelling occurred in varying degrees postoperatively [26,27]. Our study proved that PM swelling did occur postoperatively and generated complications, to a higher degree in the Moro A group than in the Moro I and II group. In addition, not only did postoperative PM swelling lengthen the PMSD and PMTD, and shrink the OW, but it also led to a short period of PM weakness. This was because PM muscles in the Moro A group required harder force and larger angle retraction to acquire a wider surgical view, resulting in more muscle traction injury and more intraoperative blood loss. A more swollen PM led to a longer PMSD and a narrower OW, which was related to more severe complications and worse clinical outcomes, as confirmed by our correlation analysis.

Previous studies have found the PM variant, known as the teardrop-shaped or high-rising psoas [29]. Voyadzis et al. concluded that a patient with a teardrop-shaped PM was not suitable for XLIF, which might increase the risk of nerve injury [30]. Whether this shape of PM affects the clinical efficacy after OLIF has not been reported yet. In our study, 13 cases of teardrop-shaped PM were identified, 92.31% of which were distributed in the Moro A group, and these patients showed a worse short-term postoperative outcome. We believed that patients whose PM was located in Moro Zone A, and especially patients whose PM had a teardrop shape, needed extra attention paid to gentle intraoperative handling and immediate postoperative anti-inflammatory and dehydration treatment. This also suggested that special attention should be paid to the teardrop-shaped PM during the preoperative imaging analysis for early intervention. We thus recommended using the Moro zone to evaluate the position and morphology of the PM preoperatively.

First, this was a retrospective clinical study and inevitably had some potential selection bias. Second, there were unavoidable errors in patient imaging measurements. Third, the study only included cases at the L4-5 level and had a small sample size and a relatively short follow-up period. Therefore, prospective studies with larger sample sizes, better designs, and longer-term follow-ups should be performed in the future.

## 5. Conclusions

The Moro zone of the PM affected the short-term outcome after OLIF. Patients whose PM was located in Moro Zone A had a longer PMSD and a narrower OW, which related to worse short-term postoperative clinical scores. Meanwhile, they showed higher estimated blood loss, higher postoperative complication rates, and a higher proportion of teardrop-shaped PM.

## Figures and Tables

**Figure 1 jcm-12-00989-f001:**
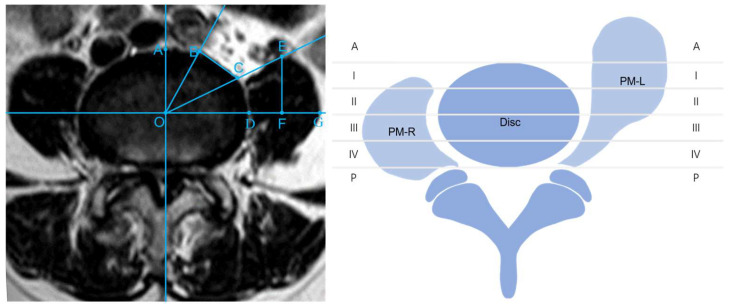
Measurement standards and Moro zones. (**left**): Disc midpoint (O); disc sagittal diameter, DSD (OA); operating window, OW (BC); psoas major transverse diameter, PMTD (DG); psoas major sagittal diameter, PMSD (EF). (**right**): The disc was divided into Zones A, I, II, III, IV, and P s. The psoas major muscles were divided according to the positions of their anterior borders. The left psoas major with a teardrop shape was located in Moro Zone A. The right psoas major with a normal shape was located in Moro Zone I.

**Figure 2 jcm-12-00989-f002:**
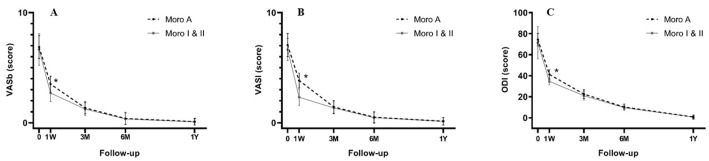
Clinical scores at different follow-up times. The VASb (**A**), VASl (**B**), and ODI (**C**) scores showed worse results in the Moro A group at 1-week postoperative follow-up. *, *p <* 0.001.

**Figure 3 jcm-12-00989-f003:**
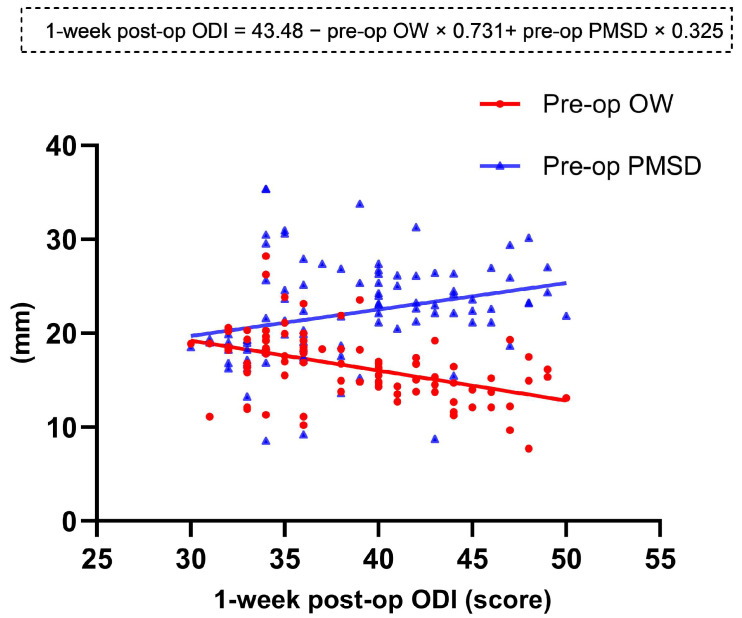
The multiple-regression linear model. The preoperative OW and PMSD showed significant linear correlations with the 1-week postoperative ODI.

**Figure 4 jcm-12-00989-f004:**
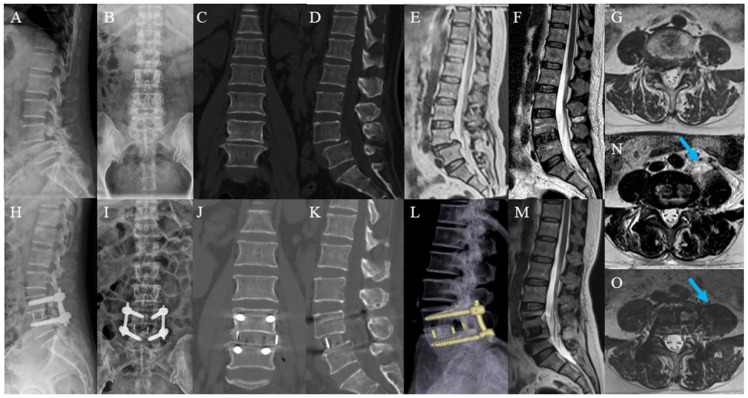
Typical case. A 55-year-old woman, presented with pain and numbness in the right lower extremity for 5 years; her condition worsened for 3 weeks. Preoperative lumbar X-ray (**A**,**B**); preoperative lumbar CT (**C**,**D**); preoperative lumbar MRI. Preoperative images showed the L4 spondylolisthesis and stenosis at L4-5 (**E**,**G**); postoperative lumbar X-ray showed L4 was reduced (**H**,**I**); postoperative lumbar CT showed the distraction of disc height (**J**,**K**); postoperative lumbar MRI showed satisfactory decompression and a huge swelling of the left psoas major (arrow) (**F**,**N**). Lumbar CT reconstruction and MRI at 1-year follow-up showed solid fusion with no cage subsidence (**L**). Swelling of the left psoas major reduced significantly and its area recovered (arrow) (**M**,**O**).

**Table 1 jcm-12-00989-t001:** Summary of patient demographics.

Parameter	Moro A Group	Moro I and II Group	*p*-Value
No. of patients	57	37	
Age at surgery in years	60.04 ± 10.03	60.68 ± 9.13	0.755
No. of M/F	21/36	10/27	0.323
BMD in g/cm^3^	−2.03 ± 0.83	−2.00 ± 0.80	0.844
BMI in kg/m^2^	23.65 ± 2.59	23.66 ± 3.09	0.989
Symptom duration in years	4.92 ± 4.53	6.22 ± 6.43	0.291
Background diseases			
CVD	13 (22.81)	8 (21.62)	0.893
Diabetes	1 (1.75)	2 (5.41)	0.559
CVD and Diabetes merge	5 (8.77)	0 (0)	0.153
Long-term smoking	1 (1.75)	1 (2.70)	1.000
Main diagnosis			
Discogenic lumbago	7 (12.28)	3 (8.11)	0.735
Spondylolisthesis, instability	21 (36.84)	13 (35.14)	0.376
Lumbar spinal stenosis	29 (50.88)	21 (56.76)	0.577

Numerical variables are expressed as mean ± SD; categorical variables are expressed as no. (%); SD: standard deviation; BMD: bone mineral density; BMI: body mass index; CVD: cardiovascular disease.

**Table 2 jcm-12-00989-t002:** Perioperative results.

Parameter	Moro A Group	Moro I & II Group	*p*-Value	
Estimated blood loss in mL	64.74 ± 25.64	54.86 ± 17.25 ^†^	0.043	
Operation time in minutes	36.09 ± 5.62	37.14 ± 6.25	0.401	
Length of hospital stay in days	8.54 ± 1.55	8.49 ± 1.68	0.865	
Complications				
Total complication rate	26 (45.61)	8 (21.62)	0.560	
Cage displacement	9 (15.79)	3 (8.11)	0.353	
Cage subsidence/endplate injury	3 (5.26)	3 (8.11)	0.677	
Thigh pain/numbness, hip flexion weakness	12 (21.05)	2 (5.41) ^†^	0.042	
Sympathetic chain injury	2 (3.51)	0	0.518	

Numerical variables are expressed as mean ± SD; complications are expressed as no. (%); ^†^ *p <* 0.05, compared with the MORO A group.

**Table 3 jcm-12-00989-t003:** Imaging parameters.

Parameter	Moro A Group		Moro I & II Group	
	Pre-Op	1-Week Post-Op	Pre-Op	1-Week Post-Op
DSD in mm	21.18 ± 1.98		22.11 ± 2.10	
OW in mm	15.81 ± 3.74	15.52 ± 3.69 ^‡^	17.45 ± 3.01 ^†^	17.36 ± 3.00 ^†‡^
PMSD in mm	25.17 ± 3.92	29.64 ± 4.79 ^‡^	17.51 ± 3.34 *	20.22 ± 3.90 *^‡^
PMTD in mm	29.56 ± 6.11	35.42 ± 6.10 ^‡^	29.42 ± 6.30	31.71 ± 5.09 ^§ψ^
Teardrop-shaped	12		1 ^†^	

Pre-op: preoperative; Post-op: postoperative; DSD: disc sagittal diameter; OW: operating window; PMSD: psoas major sagittal diameter; PMTD: psoas major transverse diameter; ^†^, *p <* 0.05, compared with the Moro A group; ^§^, *p <* 0.01, compared with the Moro A group; *, *p <* 0.001, compared with the Moro A group; ^ψ^, *p <* 0.05, compared with Pre-op; ^‡^, *p <* 0.001, compared with Pre-op.

**Table 4 jcm-12-00989-t004:** Pearson correlation analysis between pre-op imaging indicators and post-op clinical outcomes.

Parameter	1-Week Post-Op VASb	1-Week Post-Op VASl	1-Week Post-Op ODI
Pre-op DSD	−0.093	−0.132	−0.199
Pre-op OW	−0.014	−0.018	−0.465 *
Pre-op PMSD	0.422 *	0.621 *	0.276 ^§^
Pre-op PMTD	−0.021	0.076	0.012

VASb: visual analogue scale of back; VASl: visual analogue scale of leg; ODI: Oswestry disability index; Pre-op: preoperative; Post-op: postoperative; DSD: disc sagittal diameter; OW: operating window; PMSD: psoas major sagittal diameter; PMTD: psoas major transverse diameter; Equation of multiple linear regression: 1-week post-op ODI = 43.48 − β_1_ × 0.731 + β2 × 0.325 (r^2^ = 0.325).; β_1_: Pre-op OW; β_2_: Pre-op PMSD; ^§^, *p <* 0.01; *, *p <* 0.001.

## Data Availability

All data are incorporated into the article.

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
