# Peer review of "Different Moro Zones of Psoas Major Affect the Clinical Outcomes after Oblique Lumbar Interbody Fusion: A Retrospective Study of 94 Patients"

_jcm, 2023, doi:10.3390/jcm12030989_

Round 1
Reviewer 1 Report
This study is represents that OLIF for high rising psoas is prone to postoperative complications such as thigh pain.
I think this is a good study that confirms the results expected from morphology.
I have a few questions.
Is the unit of the symptom duration in years? I think it is too long if the average age is 60 years and the duration of the disease is 5 years.
2. Please provide definitions of thigh pain, hip muscle weakness, and sympathetic chain injury.
Author Response
Response to Reviewer 1 Comments
Point 1:Is the unit of the symptom duration in years? I think it is too long if the average age is 60 years and the duration of the disease is 5 years.
Response: Thank you very much for your comments. Yes, the unit of the symptom duration was indeed in years. The beginning point we recorded was the time when the patients first appeared low back pain or spinal nerve root symptoms, and the endpoint was the day before the operation. Due to the symptoms of most patients were not continuous but intermittent, after the first onset of symptoms, they usually received conservative treatment. Once the symptoms were relieved, they would be discharged, moving in circles. So the symptom duration seemed too long. But in fact, the courses of the last visit before surgery were often within 1 year.
Point 2:Please provide definitions of thigh pain, hip muscle weakness, and sympathetic chain injury.
Response: Thank you very much for your comments. Thigh pain refers to the pain on the anterior and medial surfaces of the thigh. Hip muscle weakness refers to the adynamia of the psoas major and quadriceps femoris muscle. The occurrences of thigh pain and hip muscle weakness were related to the swollen psoas major caused by intraoperative retraction and stimulation of the femoral nerves. Sympathetic chain injury refers to sympathetic chain syndrome, namely the signs and symptoms caused by the sympathetic trunk or nerve injury. Since the sympathetic trunk and nerves were distributed on both sides of the vertebrae, they might be stimulated during the process of retracting the abdominal aorta and psoas major, exposing the intervertebral space and cleaning the disc, which was likely to lead to the paresthesia or diffused pain.
Reviewer 2 Report
The authors investigated relationship the anatomy of psoas major and intervertebral disc, including Moro zone, with clinical findings in the patients underwent oblique lumbar interbody fusion (OLIF). They found that Moro zone A related to worse short term postoperative clinical scores, and there were higher blood loss, higher postoperative complication rates in patients with teardrop-shaped psoas major.
The data is interested to orthopedic and spinal surgeons. There are several concerns. I would like to advise the authors to include these points as new paragraphs in the manuscript.
1. Please spell out PM in line 42.
2. Is operating time during only OLIF without PPSF? I recommend the author describe clearly in the section materials and methods.
3. The authors showed both visual analogue scale of back and leg were correlated with PMSD. I agree with association b/w VAS of leg and PMSD, on the other hand it seems unclear what makes back pain worsening. Could you explain about that?
Author Response
Response to Reviewer 2 Comments
Point 1:Please spell out PM in line 42.
Response: Thank you very much for your suggestions. We have revised the manuscript.
Point 2:Is operating time during only OLIF without PPSF? I recommend the author describe clearly in the section materials and methods.
Response: Thank you very much for your comments. Yes, the operating time we recorded did not include posterior percutaneous screw fixation (PPSF) because the psoas major (PM) needed to be retracted only in the oblique corridor. The part of PPSF was not relevant to our research about the Moro zones of the PM. Thanks for your reminding. We have added descriptions to the section materials and methods.
Point 3:The authors showed both visual analogue scale of back and leg were correlated with PMSD. I agree with association b/w VAS of leg and PMSD, on the other hand it seems unclear what makes back pain worsening. Could you explain about that?
Response: Thank you very much for your comments. The femoral nerves emerge from the lateral border of the PM muscles, innervating the anterior and medial surfaces of the thigh. Due to a greater extent of intraoperative retraction, the PM muscles located in Moro zone A, correlating with longer PMSD, were more swollen than those in Moro â… &â…¡ after surgery, which might compress and stimulate the femoral nerves greatly. Therefore, the worse visual analogue scale of leg was correlated with longer PMSD. Similarly, swollen PM muscles might cause compression and stimulation of the paravertebral sympathetic nerves, which was likely to lead to the diffused low back pain that extended from the affected innervation area to the adjacent area. Besides, the local edema and inflammation could also make the low back pain worse. In summary, patients with longer PMSD showed worse VASl, VASb and ODI scores in the short term after surgery.